# Uncovering Limitations in Text-to-Image Generation: A Contrastive Approach with Structured Semantic Alignment

**Qianyu Feng**
MatrixVerse
University of Newcastle
qianyufeng718@gmail.com

**Yulei Sui**
University of New South Wales
y.sui@unsw.edu.au

**Hongyu Zhang**[*]
Chongqing University
hyzhang@cqu.edu.cn

## Abstract

Despite significant advancement, text-to-image generation models still face challenges when producing highly detailed or complex images based on textual descriptions. In this work, we propose a Structured Semantic Alignment (SSA) method for evaluating text-to-image generation models. SSA focuses on learning structured semantic embeddings across different modalities and aligning them in a joint space. The method employs the following steps to achieve its objective: (i) Generating mutated prompts by substituting words with semantically equivalent or nonequivalent alternatives while preserving the original syntax; (ii) Representing the sentence structure through parsing trees obtained via syntax parsing; (iii) Learning fine-grained structured embeddings that project semantic features from different modalities into a shared embedding space; (iv) Evaluating the semantic consistency between the structured text embeddings and the corresponding visual embeddings. Through experiments conducted on various benchmarks, we have demonstrated that SSA offers improved measurement of semantic consistency of text-to-image generation models. Additionally, it unveils a wide range of generation errors including under-generation, incorrect constituency, incorrect dependency, and semantic confusion. By uncovering these biases and limitations embedded within the models, our proposed method provides valuable insights into their shortcomings when applied to real-world scenarios.

## 1 Introduction

Text-to-image generation models have made significant progress over the past few years, leveraging advancements in deep learning and natural language processing techniques. These models aim to generate visually coherent and semantically relevant images from textual descriptions, pushing the boundaries of generative techniques. Compared

---

[*]Corresponding author.

to the visual clues (segmentation maps, regional edges) adopted for single-modal image synthesis, language provides an alternative but more intuitive description. The progress in text-to-image generation models has opened up possibilities in various domains, including content creation, virtual worlds, and entertainment. The continued advancements in deep learning techniques, combined with larger and more diverse datasets, hold promise for even more realistic and context-aware image synthesis from text in the future. The journey began with early attempts using convolutional neural networks (CNNs) and recurrent neural networks (RNNs). However, these models struggled to capture the complex relationship between text and images. With the introduction of generative adversarial networks (GANs), the field witnessed a major breakthrough. Progress continued with models such as DALL-E (Ramesh et al., 2021) and CLIP (Radford et al., 2021) leveraged reinforcement learning to generate images from textual prompts. DALL-E demonstrated the ability to generate highly creative and unique images based on textual input, while CLIP enabled image generation conditioned on text descriptions. Diffusion models have emerged as a powerful approach for image generation, capable of generating high-quality and diverse images. These models leverage the principles of diffusion processes to iteratively transform a noise vector into a realistic image. Diffusion models can generate diverse and high-resolution images with rich details. They also provide a fine-grained level of control over the generated images by conditioning on specific inputs.

However, the effective transfer and fusion of heterogeneous information from different modalities in text-to-image generation tasks remain a significant challenge due to substantial domain gaps. Moreover, there are often many-to-many mapping relationships, meaning that one image may correspond to multiple textual descriptions, and vice

versa. As a result, the generated images may not always align with the text prompts provided by users, requiring manual validation and selection of satisfactory results. Various aspects of the generated images, such as visual quality, object accuracy, attributes, and contextual information, need to be assessed. Therefore, an automatic evaluation process and comprehensive and objective evaluation metrics are crucial for evaluating the effectiveness of text-to-image generation models. However, evaluating these models poses challenges as the generation process may not always be stable, and the perception of image quality is often subjective. Currently, the primary factors considered in the evaluation process are image quality and text-image similarity. Popular metrics include Inception Score (IS) (Salimans et al., 2016) and Frechet Inception Distance (FID) (Heusel et al., 2017) for assessing the image fidelity, and R-precision (Xu et al., 2018) and CLIP score (Radford et al., 2021) for measuring the cross-modal alignment. However, the existing metrics lack insights into the assessment of fine-grained semantic concepts, such as object attributes, context details, and semantic relationships. These factors are critical in evaluating the performance of text-to-image generation models, especially with complicated input prompts.

In this context, we introduce a novel approach called Structured Semantic Alignment (SSA) to assess the accuracy and robustness of text-to-image generation models. SSA aims to evaluate the successful transfer of joint semantics from text prompts to the generated images. The method involves generating similar prompts by replacing one word in a given sentence with a semantically similar and syntactically equivalent word. Images are then generated using these prompt pairs. The underlying concept of SSA is that a reliable text-to-image generation model should consistently produce coherent and aligned results when given prompts with similar meanings. Conversely, the model should generate distinct outputs when presented with prompts containing different semantic concepts. Thus, when there is a mutation in the prompt, the generated image should still align with the input, indicating the model's robustness. This allows for the evaluation and diagnosis of the model's performance under varying conditions. SSA consists of the following key procedures:

(1) Mutated prompt generation: We first generate source prompt mutations by replacing a word in

the original sentence with a semantically equivalent or nonequivalent word with similar syntax. This variation enables the exploration of different inputs.
(2) Structured embedding learning: The sentence is represented using syntax parse trees with constituency or dependency parsing. This approach breaks down the input sentence into constituent parts, facilitating the understanding of its structure and meaning.
(3) Cross-modal feature fusion: SSA automatically creates a scene graph by parsing the image. The scene graph is a hierarchical representation of objects and their attributes, as well as their spatial relationships. Nodes represent objects, while edges represent relationships between them.
(4) Contrastive semantic alignment: Lastly, SSA integrates multi-modal contrastive learning into the feature processing fusion module, which encodes similar representations for positive pairs and different representations for negative pairs.

The implementation of the SSA methodology allows for a comprehensive evaluation of text-to-image generation models, specifically their ability to accurately transfer semantics from text prompts to the generated images. This approach serves as a valuable tool for diagnosing and enhancing the performance of such models, ensuring their reliability across diverse semantic contexts. Moreover, SSA enables the identification of cases where the generated images significantly deviate from the given prompt, enabling the assessment of accuracy and robustness in text-to-image generation models. Additionally, we conduct a thorough evaluation of the current state-of-the-art methods for text-to-image models, utilizing SSA to identify potential weaknesses in these models.

## 2 Challenges

Text-to-image generation models have made significant progress in recent years, but they still have some limitations and under-performance. The generation is heavily dependent on the quality and quantity of training data. If the training data is biased or limited, the generated images will show the contents dis-aligned with text. This problem can also be attributed to the complexity of image generation models, which rely on deep neural networks and complex optimization algorithms to generate images. These models often have a limited understanding of the semantics of the given context and may struggle to maintain semantic consistency

while generating images.

**Fine-grained semantic consistency.** Conditional image generation models are designed to generate images based on some given context or condition, such as a text description or a style image. However, these models often face the problem of limited ability in semantic consistency. The models can struggle to present accurate content aligned with the text description, especially when dealing with rare concepts or complex scenarios. Ideally, if a model is given a text description and is tasked with generating an image based on this description, it should generate images consistent with the given context. However, due to the model's limited ability to maintain semantic consistency, it may generate an image that deviates from the given context. Examples in Fig. 1 show some failures in the generation of details. In the right two images, models generate "an orange car" given "an orange near a car".

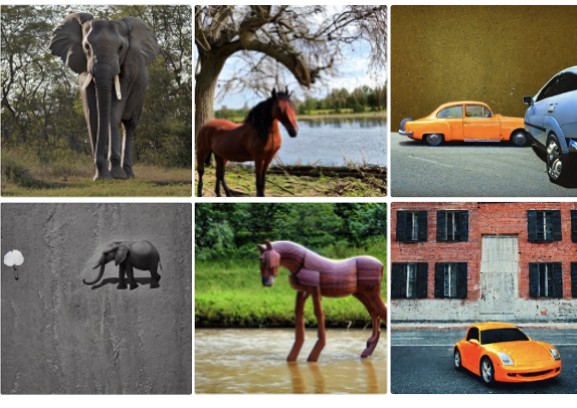

Figure 1: Images generated by Stable Diffusion given "an elephant in the air", "a wooden horse stands by a river", and "an orange near a car".

**Data biases and ambiguity.** Generating high-quality images with fine-grained details can be challenging for text-to-image models. This can lead to generating images that are unnatural or lack important details. Fine-grained details can include subtle textures, patterns, colors, and shapes that are unique to the objects or scenes described in the text. Another challenge is balancing the generation of fine-grained details with overall image quality. Generating high-quality images with fine-grained details requires the model to accurately represent the entities and their context from the input description, which can be a difficult task, especially when dealing with abstract concepts or complex descriptions.

**Difficulty in handling long and complex prompts.** Text-to-image models usually fail to generate images of complex scenes with multiple objects or tokens with multiple semantic meanings. This can lead to unrealistic or inaccurate results. Complex scenes may include natural scenes, urban environments, or indoor scenes, among others. These scenes require the model to understand the spatial relationships and interactions between the objects and the environment, as well as other attributes such as textures, lighting, and colors. Largely relying on the deep learning of different modalities, models can map the visual content with the wrong text embeddings.

## 3 Method

### 3.1 Overview

In this study, the proposed Structured Semantic Alignment (SSA) model consists of three essential modules as illustrated in Fig. 2. During the generation process, it is observed that the generated images may contain certain details that are not explicitly described in the input prompt. To ensure consistency and evaluate the alignment of corresponding semantics across modalities, we first extract the semantic components from the input prompt using syntax parsing, and from the generated image using a scene graph parser (Tang et al., 2020). This approach enables us to identify and match relevant semantic elements between the text and image representations. Following that, we employ graph attention layers to integrate node embedding from its neighbors.

In the second module, referred to as Cross-modal Feature Fusion, we present how to integrate multi-modal semantics into a joint space. Furthermore, attention layers dynamically assign weights to each modality, enabling the formation of a comprehensive embedding representation. This module enhances the precision and efficiency of semantic alignment, leading to a more effective fusion of multi-modal knowledge. Finally, through contrastive learning, we continuously compare negative and positive sample sets to obtain high-confidence modality embeddings. The mutated prompts are generated by substituting words with semantically equivalent or nonequivalent alternatives while preserving the original syntax.

### 3.2 Structured Embeddings

Given the similarity in structures between aligned semantics in different modalities, the graph struc-

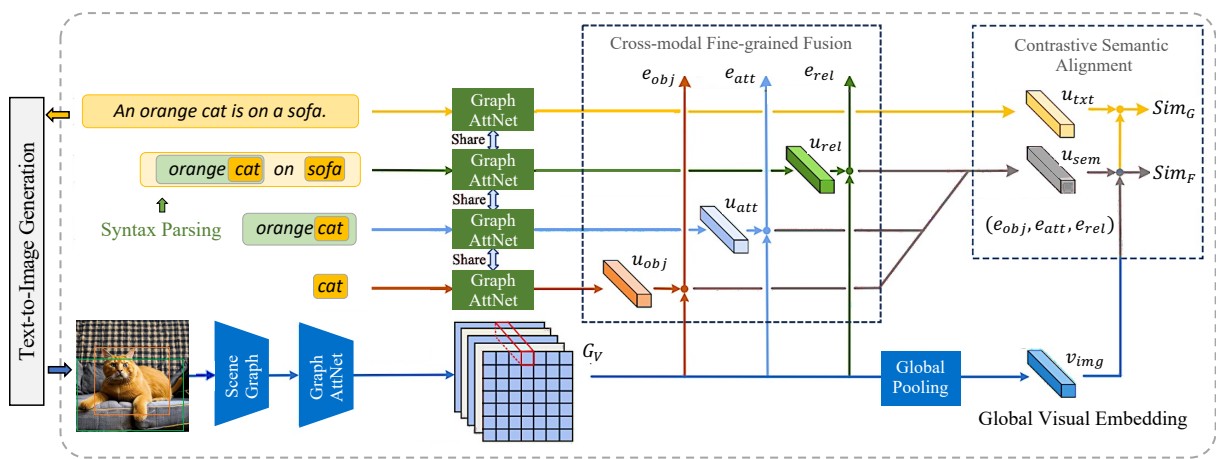

Figure 2: The framework of Structured Semantic Alignment (SSA) consists of three modules: (1) Structured embeddings learning by first parsing the text syntax and the generated image into a scene graph and applying graph attention mechanism; (2) Cross-modal Feature Fusion with cross-attention applied on the node embeddings; (3) Contrastive Semantic Alignment encodes similar representations for positive pairs and different representations for negative pairs.

ture information becomes valuable for multi-modal tasks. In this paper, graph attention networks are employed to effectively model the structural information of both $G_T$ derived from the syntax tree of input text and $G_V$ from the scene graph of generated images. This allows for the direct integration of graph-based representations to enhance the alignment process. Each node aggregates the hidden states of its neighbors $N$, through self-loops, represented as:

$$e_i^g = \|_{k=1}^{K} \sigma \left( \sum_{j \in N_i} a_{ij}^k h_j \right),\qquad (1)$$

where $h_j$ is the hidden state of node $j$. $\sigma$ denotes the ReLU non-linear operation, and where $a_{ij}^k$ is the normalized attention coefficient obtained from the $k_{th}$ attention calculation with splicing operation $\|$. $K$ is the number of attention heads. $a_{ij}$ also represents the importance of node $j$ to node $i$, calculated through self-attention:

$$a_{ij} = \frac{\exp \left( \sigma \left( a^T [Wh_i \oplus Wh_j] \right) \right)}{\sum_{u \in N_i} \exp \left( \sigma \left( a^T [Wh_i \oplus Wh_u] \right) \right)}. \quad (2)$$

### 3.3 Fine-grained Semantic Alignment

The feature processing and fusion module encompasses two aspects of work. Firstly, contrastive learning is employed to enhance the representation of intra-modality in the feature processing fusion module. This process strengthens the feature representation within each modality and captures the

intra-modality dynamics, which establish discriminative boundaries for each modality in the embedding space.

To facilitate this, a multi-modal contrastive learning module is integrated into the feature processing fusion module, incorporating contrastive loss. Positive and negative samples are created for each modality, and contrastive learning is performed to minimize the loss. The paper employs a loss formula that encodes similar representations for positive pairs and different representations for negative pairs, as shown below:

$$L\left(E, E'\right) = \frac{1}{2N} \sum_{n=1}^{N} Y d^2 \left(e_i, e_i'\right) + \\ (1 - Y) \max \left(\delta - d\left(e_i, e_i'\right), 0\right)^2, \qquad (3)$$

where $Y$ represents the label of pairs, $d$ denotes the cosine similarity, $N$ indicates the number of batch samples, $\delta$ denotes the margin hyper-parameter. $e_i \in E$ and $e_i' \in E'$ represent the corresponding semantic in two graphs $G_T$ and $G_V$. The final score of SSA is the sum of the global similarity $Sim_G = d(u_{txt}, v_{img})$ and the fine-grained semantic similarity $Sim_F$ between $k$ pairs of text embeddings and image embeddings:

$$SSA = d(u_{txt}, v_{img}) + \frac{1}{k} \sum_{i=1}^{k} d(u_{s_i}, e_{s_i}). \quad (4)$$

### 3.4 Contrastive Prompt Transformation

The underlying principle of the proposed SSA is that an ideal text-to-image generation model should

produce similar outputs when provided with similar prompts, while generating distinct results when the semantics of the input prompt change. To accomplish this, the procedure of prompt mutation generation is as follows: (1) Given an input prompt, the sentence is parsed into a syntax tree. (2) A semantic token (such as a noun, adjective, or preposition) is selected from the text to be replaced. (3) A random selection is made for a semantic equivalent or inequivalent word, which modifies the prompt and creates a mutated version.

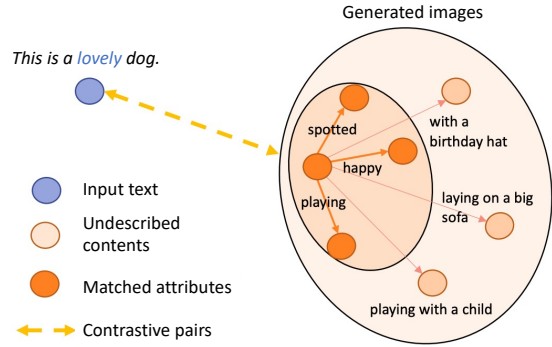

Figure 3: Contrastive learning with positive and negative pairs aligns the corresponding semantic embeddings.

To achieve this mutation, we alter a single token in the input sentence, resulting in an alternative sentence that maintains the same syntactic structure. This approach generates a set of sentences that possess identical structures while being semantically similar. Subsequently, new images are generated using the mutated prompts, and the cross-modal consistency between the synthesized images and the mutated prompts is examined. An accurate multi-modal synthesis model should generate images with comparable semantic content when provided with similar prompts, while producing different results when given different prompts.

The motivation behind this heuristic is that the constituents of a sentence should stay the same between two sentences where only a single token of the same part of speech differs. In a robust conditional image generation system, this should be reflected in the target images as well. Hence, we calculate the similarity between the syntax tree $G_T$ of the input sentence and the scene graph $G_V$ of the generated image by accumulating the distances of nodes and edges. Again, the motivation is that the structured semantic embedding will ideally remain unchanged when the token is replaced with a

similar word or phrase. Therefore, a change in the set is a reasonable indication that structural invariance has been violated and presumably there is a generation error.

# 4 Experiment

## 4.1 Experimental Settings

**Models** In this section, we conduct experiments to evaluate our evaluation method on the popular and open-sourced text-to-image generation models including: (1) GAN-based models pre-trained on MS COCO: StackGAN (Zhang et al., 2017) generates high-resolution images by stacking two GANs, and DF-GAN (Tao et al., 2020), which enhances the text-image semantic consistency with a target-aware discriminator; (2) DALL-Es: We experiment with two open-sourced implementations of DALL-E by (Dayma et al., 2021), i.e., Dall-E mini [1] and Dall-E mega [2]. (3) Diffusion-based models: Composable Diffusion (Liu et al., 2022) which introduces compositional operators with multiple diffusion models[3], and Stable Diffusion (Rombach et al., 2022) which is a latent text-to-image diffusion model using a frozen CLIP ViT-L/14 text encoder to condition the model on text prompts and trained a subset of LAION-5B (Schuhmann et al., 2022).

**Evaluation Metrics** We evaluate our method using the widely-used metrics in the experiments. IS (Salimans et al., 2016) is a metric that measures the quality and diversity of generated images. It is based on a pre-trained inception model that is used to classify images. Higher IS scores indicate that the generated images are both high quality and diverse. FID (Heusel et al., 2017) is a metric that measures the distance between the distributions of the generated images and real-world images. It is based on a pre-trained inception model that is used to extract feature vectors from images. Lower FID scores indicate that the generated images are closer to real-world images in terms of their visual content and quality.

CLIP (Hessel et al., 2021) is a metric that measures the similarity between generated images and real-world images. It is based on a pre-trained

---

[1]Dall-E mini: https://huggingface.co/dalle-mini/dalle-mini

[2]Dall-E mega: https://huggingface.co/dalle-mini/dalle-mega

[3]https://colab.research.google.com/github/energy-based-model

Table 1: Quality metrics evaluating text-to-image generation models on mutated prompts from MS COCO.

| Model | IS↑ | FID↓ | R↑ | CLIP↑ | SSA↑ |
|---|---|---|---|---|---|
| StackGAN | 10.1 | 27.2 | 46.3 | 18.7 | 42.7 |
| DF-GAN | 11.3 | 22.4 | 50.6 | 20.5 | 53.6 |
| Dall-E mini | 14.6 | 17.8 | 63.1 | 23.2 | 62.5 |
| Dall-E mega | 19.9 | 14.6 | 69.5 | 25.3 | 65.8 |
| Stable Diffusion | **24.5** | **9.4** | 70.8 | **36.2** | 72.1 |
| Composable Diffusion | 24.1 | 11.2 | **71.3** | 33.7 | **76.6** |

Table 2: Evaluation metrics for models on DrawBench.

| Model | R↑ | CLIP↑ | SSA↑ |
|---|---|---|---|
| StackGAN | 44.5 | 16.2 | 34.6 |
| DF-GAN | 53.4 | 18.4 | 49.0 |
| Dall-E mini | 60.7 | 23.2 | 52.8 |
| Dall-E mega | 62.0 | 25.3 | 55.9 |
| Stable Diffusion | 67.8 | **32.5** | 62.7 |
| Composable Diff. | **69.4** | 31.2 | **66.3** |

model that can recognize visual features and associate them with words, allowing it to judge whether an image is semantically meaningful or not. Higher CLIP scores indicate that the generated images are more similar to real-world images. R precision (Xu et al., 2018) is a score that learns the embeddings of real images that are similar to the embeddings of their corresponding captions. To evaluate a generative model, we use one of the captions to generate an image from it. We then sample another 29 captions randomly from the dataset and calculate the similarity between the embedding of the generated image and the embeddings of the 30 captions.

**Datasets** We first evaluate our method on the evaluation set of MS COCO (Lin et al., 2014). We use 250 randomly chosen image-caption pairs from the validation set. Specifically, we generate 5 syntax-similar prompt mutations per sentence. In total, we have 1500 prompts for testing the text-to-image generation models. DrawBench (Ruiz et al., 2023) is a dataset for a multi-dimensional evaluation of text-to-image models, with text prompts designed to probe different semantic properties of models. It contains 11 categories of prompts, testing different capabilities of models such as compositionality, cardinality, spatial relations, and the ability to handle complex text prompts or prompts with rare words.

## 4.2 Evaluation Results

Most text-to-image works evaluate with metrics for the quality and fidelity of the generated images, *e.g.*, FID and IS, which barely reveal the semantic consistency. Differently, CLIP and R measure the similarity between the embeddings of text prompts and generated images, but they are limited in verifying fine-grained semantics. Comparatively, SSA better measures the consistency of semantic concepts and provides more insights into fine-grained alignment. The metrics evaluated on MS COCO are reported in Table 1. Furthermore, we compare the semantic metrics CLIP, R, and SSA on a more challenging benchmark DrawBench, as detailed in Table 2. The findings demonstrate that stable diffusion excels in producing images with superior fidelity. In contrast, composable diffusion exhibits a stronger capacity to capture fine-grained semantic nuances, as evidenced by its higher R and SSA scores. It is noteworthy that SSA proves to be less susceptible to the quality of generated images, and it provides deeper insights into the intricacies of fine-grained semantic consistency.

In Table 3, we assess Dall-E and diffusion-based models using prompts with varying degrees of mutation sourced from MS COCO. Overall, diffusion-based models perform better than Dall-E models while Composable Diffusion generates more accurate fine-grained details than Stable Diffusion. Table 4 reveals that Stable Diffusion yields the most favorable performance when mutations affect the *subject* term, while Composable Diffusion demonstrates greater resilience to diverse mutations impacting the *relation* term. Furthermore, our evaluation indicates that Dall-E mega, which benefits from additional training data, exhibits enhanced robustness to mutated prompts. Additionally, the evaluation demonstrates that SSA is a valuable tool for gauging the fine-grained consistency in mutated prompt generation.

Table 3: Evaluation with SSA on prompts with different mutations. Percentage is calculated with mutated words over the total words in the sentence.

| Model | Dall-E mini | Dall-E mega | Stable Diffusion | Composable Diffusion |
|---|---|---|---|---|
| Original | 62.5 | 65.8 | 72.1 | 76.6 |
| Mutations (13%) | 58.6 | 62.2 | 70.4 | 74.3 |
| Mutations (29%) | 51.3 | 54.0 | 65.8 | 68.9 |
| Mutations (54%) | 47.4 | 50.5 | 62.1 | 64.7 |

Table 4: Evaluation of Dall-Es and diffusion models on original prompts and prompts with different types of mutations with SSA.

| Model | Prompt | M_sub | M_obj | M_rel |
|---|---|---|---|---|
| Dall-E mini | 62.5 | 58.2 | 52.6 | 47.8 |
| Dall-E mega | 65.8 | 65.3 | 59.9 | 54.6 |
| Stable diffusion | 72.1 | **67.2** | 64.1 | 59.4 |
| Composable diff. | **76.6** | 69.7 | **64.5** | **64.2** |

### 4.3 Discovered Typical Generation Errors

The proposed SSA is capable of discovering generation errors of diverse kinds. In our experiments with DALLE and Diffusion models, we mainly find four types of generation errors: under-generation, incorrect constituency, incorrect dependency, and semantic ambiguity. To provide a glimpse of the diversity of the uncovered errors, this section gives some examples of the errors.

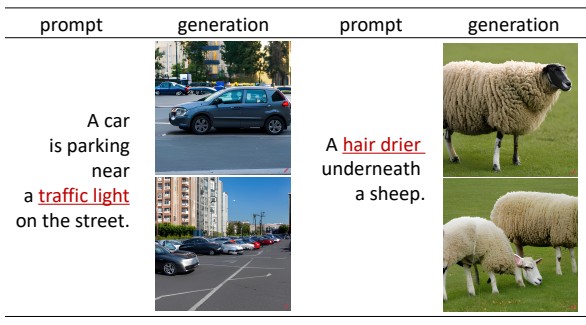

Figure 4: Under-generation error examples.

#### 4.3.1 Under-Generation

If some words are mistakenly missing (i.e. do not appear in the generated image), it is an under-generation error. Fig. 4 presents examples that contain under-generation errors. In the generation, "traffic light" and "hair drier" are mistakenly missing, which leads to the target image of different semantic content. In our observation, under-

generation errors often take place when an object is combined with a rare context. Models also fail to generate all the constituencies when the description is too long or too complicated.

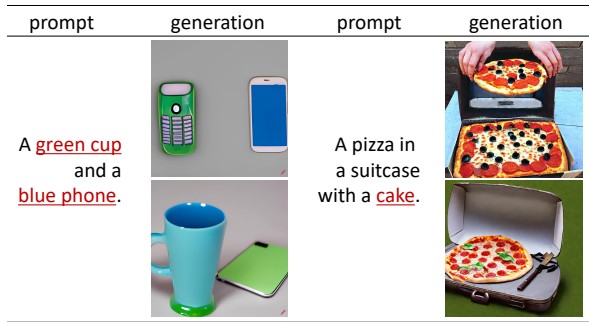

Figure 5: Examples of incorrect constituency errors.

#### 4.3.2 Incorrect Constituency

A token/phrase could also be incorrectly generated to another concept that seems semantically unrelated. For example, in Fig. 5, the image on the left-top is generated with two phones by mistake. The left-bottom image has the right objects but incorrect color attributes. In the right example, the top image contains two pizzas in a suitcase. While in the bottom image, a pizza and a tool in a suitcase are generated. Many errors of this type appear when models are in accordance with the right structure of the scene graph but fail to generate the correct objects or context.

#### 4.3.3 Incorrect Dependency

If all the constituencies are correctly generated but the relationship between them is incorrect, it is an incorrect dependency error. In Fig. 6, all the objects are generated correctly in the images. However, in the left example, the bicycle is under the boat rather than "on top of". In the right example, the cat is inside the box in the top image and on the left of the box in the other image. It is also observed that models struggle to generate the right relationship that is not prevalent in daily life.

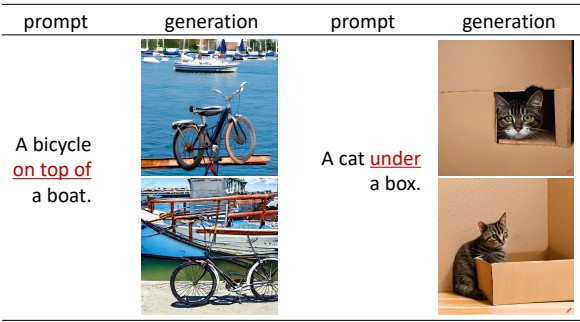

| prompt | generation | prompt | generation |
|---|---|---|---|
| A bicycle on top of a boat. | | A cat under a box. | |

Figure 6: Incorrect dependency error examples.

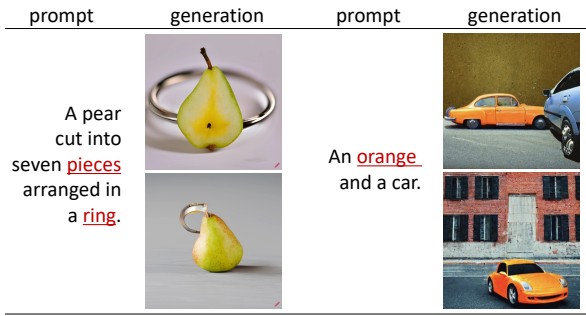

| prompt | generation | prompt | generation |
|---|---|---|---|
| A pear cut into seven pieces arranged in a ring. | | An orange and a car. | |

Figure 7: Semantic confusion generation errors.

### 4.3.4 Semantic Ambiguity

Semantic confusion refers to a state of uncertainty or ambiguity that arises when there is a lack of clarity or consistency in the meaning of terms or concepts. It can occur in various contexts, such as in language, communication, or knowledge representation, and can lead to misunderstandings or errors in reasoning. In image generation, semantic confusion can arise due to the multiple meanings of words or phrases, idioms or figurative language, or the lack of context. In Fig. 7, on the left, the model fails to understand the "ring" describes the shape of pear pieces instead of the "ring" shown in the image. On the right, "orange" should be generated as a fruit but the model confuses and generates a car in the color of "orange". These examples indicate that the current generation models need to be improved with respect to the semantic understanding of token/phrase in its context.

## 5 Related Work

**Text-to-Image Generation.** The goal of text-to-image generation is to create realistic images with textual descriptions. The prosperity of large-scale Generative Adversarial Networks (GANs) (Brock et al., 2018; Karras et al., 2019, 2020, 2021) sig-

nificantly advances research on multi-modal image synthesis. By leveraging contrastive language-image pre-training (CLIP) (Radford et al., 2021) or GAN inversion techniques (Xia et al., 2022), pre-trained GANs can be applied to text-driven image synthesis and editing tasks. Recently, the power of handling input from multiple modalities has led to the popularity of Transformer models (Vaswani et al., 2017). Significant progress has been made in multi-modal image synthesis. For example, DALL-E (Ramesh et al., 2021, 2022) trained a large-scale auto-regressive Transformer on a large amount of image-text pairs to produce a high-fidelity generative model through text prompts. Stable Diffusion (Rombach et al., 2022) adopted latent diffusion models to achieve favorable results across a wide range of multi-modal image synthesis tasks. Although high-fidelity image synthesis is significantly advanced by those efforts, the fine-grained semantic alignment between the text and the generated images is seldom deeply investigated.

**Evaluation of Text-to-Image Models.** The rapid advance in text-to-image generation offers unprecedented generation realism and editing possibilities, which have influenced and will continue to influence our society. Establishing an accurate, reliable, and systematic evaluation framework is highly necessary to evaluate text-to-image generation models. However, evaluating the quality of the image generative models has proven to be challenging, as demonstrated in (Theis et al., 2015). In most of the multi-modal image synthesis methods (Ramesh et al., 2021, 2022; Rombach et al., 2022), image quality and text-image alignment are the main factors considered in the evaluation process. Commonly used evaluation metrics are Inception Score (IS) (Salimans et al., 2016) and the Frechet Inception Distance (FID) (Heusel et al., 2017) for image fidelity, and CLIP score (Radford et al., 2021) for text-image alignment. The metric R-score (Xu et al., 2018) relies on the Cosine similarity between image and text embeddings to determine whether a generated image is more similar to the ground truth texts than random samples from the dataset. However, R-score does not directly measure semantic consistency and can be highly biased by the dataset. SOA (Hinz et al., 2020) attempts to address this issue by utilizing a pre-trained object detection model to evaluate whether an object mentioned in the text exists in the generated image. However, SOA is not suitable for datasets where

only one object appears in the generated images. CLIP score (Hessel et al., 2021), which is designed for image captioning and adopts cosine similarity of CLIP embeddings, may not explicitly associate attributes with objects and overlooks the semantic variations. Because none of the existing measures are comprehensive enough, current works utilize many metrics. The performance assessment is even more challenging in the text-to-image generation task due to the multi-modal complexity of text and image. This motivates us to develop a new evaluation method to compare text-to-image alignment.

# 6   Conclusion

The effectiveness and robustness of text-to-image generation models are of vital importance to ensure their practicality in real-world applications. In this paper, we introduce Structured Semantic Alignment (SSA) with contrastive learning for text-to-image generation models. Through our experiments, we have identified limitations of existing state-of-the-art models in text-to-image generation. Our assessment highlights that these models may struggle with synthesizing semantic concepts in a given context. Designing methods for overcoming the limitations and enhancing the model's ability to handle a wider range of semantic variations would be an intriguing area for future research. Improving the understanding and measuring the fine-grained semantic consistency across different modalities is a critical aspect to consider in advancing multi-modal generative techniques.

# 7   Limitations

While the proposed method described above offers a comprehensive evaluation approach for text-to-image generation models, it also has its limitations. Here are some limitations of this method:

*Limited Semantic Expressiveness.* The proposed SSA method relies on prompt mutations where a single word in the original sentence is replaced with a semantically similar or nonequivalent word. While this variation enables the exploration of different inputs, it may not capture the full range of semantic concepts and complexities present in natural language. As a result, the evaluation might not fully reflect the model's ability to understand and generate diverse visual interpretations based on complex textual descriptions.

*Dependency on In-domain Data Resources.* While SSA provides a systematic evaluation frame-

work, the word embeddings and scene graph construction from the pre-trained models may introduce some in-domain biases. The effectiveness of SSA relies on the expressiveness of the representation of semantic components from each modality. If the text token or generated content is out of the domain of pre-trained models, it can limit the comprehensiveness of the evaluation. Similarly, the interpretation and analysis of semantic contrastive comparison results can be subjective, potentially introducing biases in the evaluation process.

*Limited Assessment of Image Quality.* The SSA method primarily focuses on evaluating the alignment and consistency between generated images based on prompt mutations. While this provides insights into the model's robustness and ability to maintain coherence with variations in prompts, it doesn't directly assess the quality and fidelity of the generated images. Factors like visual realism, level of detail, and adherence to specific attributes or style may not be adequately captured by the semantic contrastive comparison alone. Therefore, the SSA method may not provide a holistic evaluation of the overall image generation capabilities of text-to-image models.

It is important to recognize these limitations and consider them when interpreting the results and conclusions derived from the SSA approach. Future research and refinement of SSA and similar methodologies will aim to address these limitations to ensure a more reliable and comprehensive evaluation of text-to-image generation models.

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

## A  Architecture

### A.1  Structure Embedding Extraction

**Scene Graph** is a graph-based semantic representation of image contents. They encode the objects in an image, their attributes, and the relationships between objects. Parsing an image into a scene graph involves several steps, including object detection, object recognition, and relationship extraction. A high-level overview of the process is as follows: (i) Object detection: The first step is to detect the objects in the image. Detection models can be used to identify objects and their locations. (ii) Relationship recognition: We extract the relationships between the objects in the image. This involves identifying how the objects are related to each other, such as whether one object is in front of or behind another, or whether two objects are next to each other. (iii) Scene graph construction: Once the objects and their relationships have been identified, a scene graph can be constructed. It can be represented as a directed graph, where the objects are nodes and the relationships between them are edges.

**Syntax Parsing** aims to create a structural representation of a sentence, typically in the form of a parse tree or a dependency tree. We apply the dependency parsing which focuses on identifying the relationships between words in a sentence by representing them as directed dependencies. Each word in the sentence is considered a node, and the relationships between the words are represented as directed edges. The edges indicate the syntactic role of each word and its dependence on other words. By analyzing the syntactic structure, models can identify the subject and object of a sentence, extract key phrases, detect negation or sentiment-bearing words, and perform various linguistic analyses.

### A.2  Graph Attention Network

Due to the similarity of the structures of aligned semantics across different modalities, graph structure information is utilized for multi-modal alignment tasks. The first step is to construct a graph representation of the data, where nodes represent entities or elements of interest, and edges capture the relationships or connections between them. Each node in the graph is associated with an initial node embedding. To enhance the modeling capacity and capture different aspects of the graph, GAT often employs multiple attention heads. Each attention head independently computes attention weights and

performs aggregation, providing multiple views or representations of the graph. Attribute feature embedding plays a crucial role in our work as it leverages attributes to provide information about an object's details. Our approach involves two distinct stages. Firstly, all attributes within the graph are extracted and stored first. Subsequently, two sets of operations are performed.

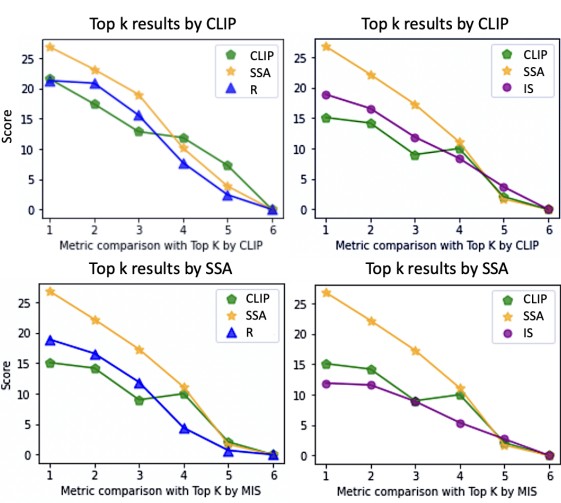

Figure 8: Different metrics evaluated on results ranked by CLIP score in the first row and results ranked by SSA in the second row.

In one set of operations, we treat object attributes similarly to object structure representation. This involves disregarding attribute values and focusing on extracting each attribute associated with an object. When aligning objects from two different knowledge graphs, we observe that the objects to be aligned often possess similar attribute structures. Exploiting this observation, we simulate the attribute structure to create a comprehensive representation. In the other set of operations, we aim to represent the object attributes in a suitable format.

In this paper, we set the embedding size for all layers to 128 and used a mini-batch method with a batch size of 512. For each experiment, the model is trained for 300 epochs and set the corresponding learning rates. In our approach, we utilize GloVE word embeddings with a hidden dimension of 300. Phrases are segmented into individual words based on spaces. Out-of-vocabulary words are replaced with a special token, <UNK>.

To obtain object proposals, we employ an off-the-shelf Faster RCNN model as the object detector. The backbone of this detector is ResNet-101, which has been pre-trained on Visual Genome. Regard-

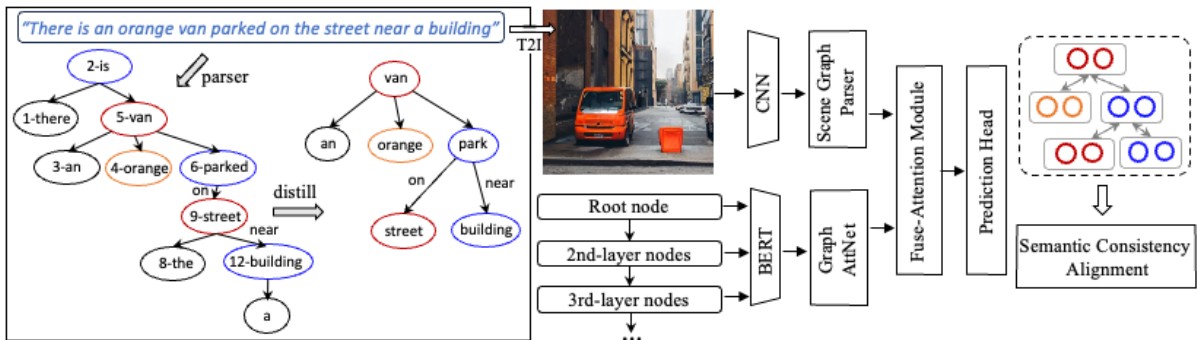

Figure 9: The pipeline of the evaluation of semantic consistency with SSA.

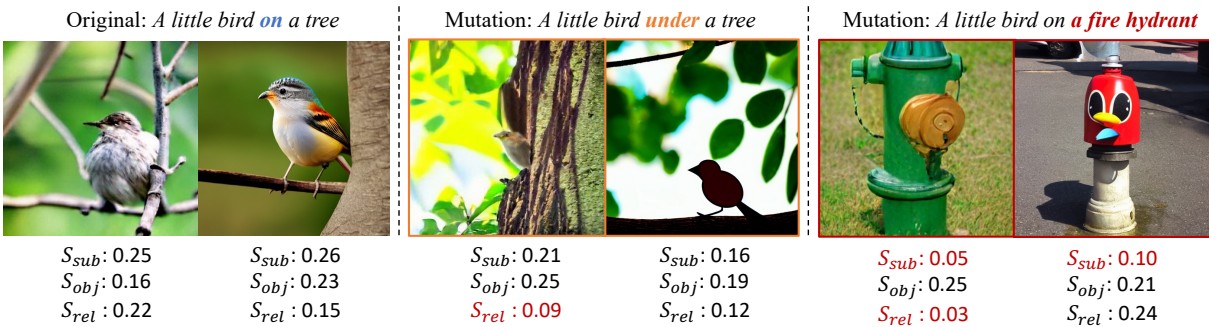

Figure 10: Example of mutations on the relation component and object component: Given "A little bird on a tree" as the input prompt, we change "on" to "under" and "tree" to "fire hydrant".

ing visual features, we use the 2048-dimensional feature vectors obtained from Bottom-up attention. These features are pre-trained using 1600 object labels and 400 attributes from the dataset. It is important to note that the visual features extracted from this process remain frozen during the training phase.

## B Prompt Mutation

To assess structural invariance, it is necessary to compare two sentences with identical syntactic structures but differing in at least one token. Our approach focuses on modifying a single token in an input sentence while adhering to certain constraints, resulting in a set of sentences that are both structurally identical and semantically similar. Specifically, we modify one token in the input sentence at a time by replacing it with another token of the same part of speech. For instance, in the source sentence, we mask the token "lovely" and replace it with the top-k most similar tokens, generating k similar sentences. This process is applied to every candidate token in the sentence, limited to nouns and adjectives to maintain grammatical correctness.

There are also challenges in selecting replacement tokens. One straightforward approach is to utilize word embeddings. This involves choosing words with high vector similarity and identical tags as replacements for the original token in the modified sentences. However, since word embeddings lack contextual information, this approach often yields sentences that do not align with common language usage.

For instance, while the word "fork" may exhibit high vector similarity and share the same POS tag as the word "plate", the sentence "He uses the plate to eat" does not make sense, unlike the coherent sentence "He uses a fork to eat".

Hence, the selection of replacement tokens must consider both vector similarity and contextual appropriateness to ensure that the generated sentences maintain linguistic coherence and align with natural language usage.

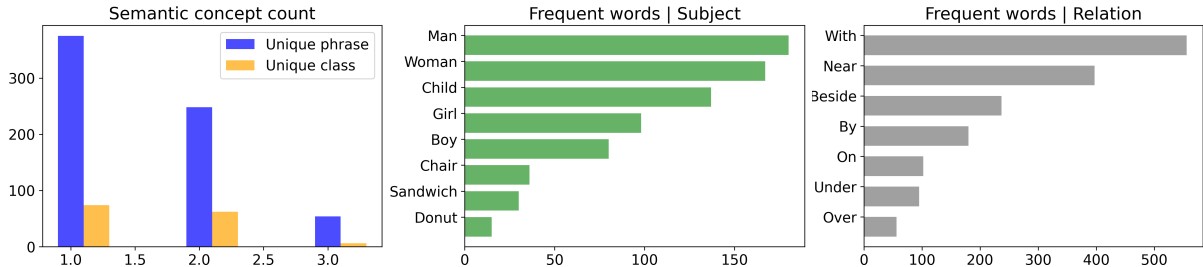

Figure 11: Statistics of semantic concepts by phrase and by category in the prompts of DrawBench. The right two plots illustrate the most frequent words that appear as "subject" and "relation" in the sentence syntax respectively.

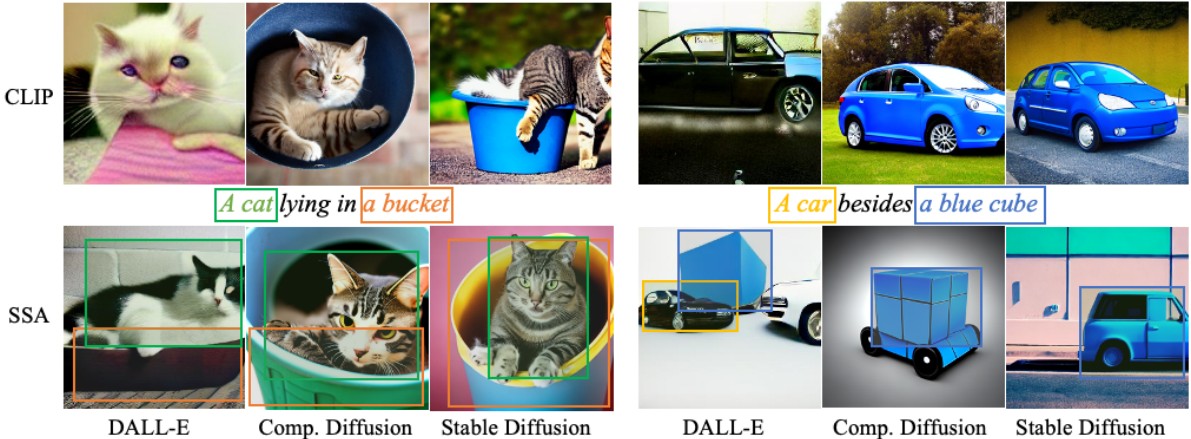

Figure 12: Comparison of generation results ranked by CLIP and SSA given prompts: (i) "A cat lying in a bucket" and (ii) "A car besides a blue cube". The green and orange bounding boxes in the prompts and the images indicate the fine-grained alignment of semantic concepts.

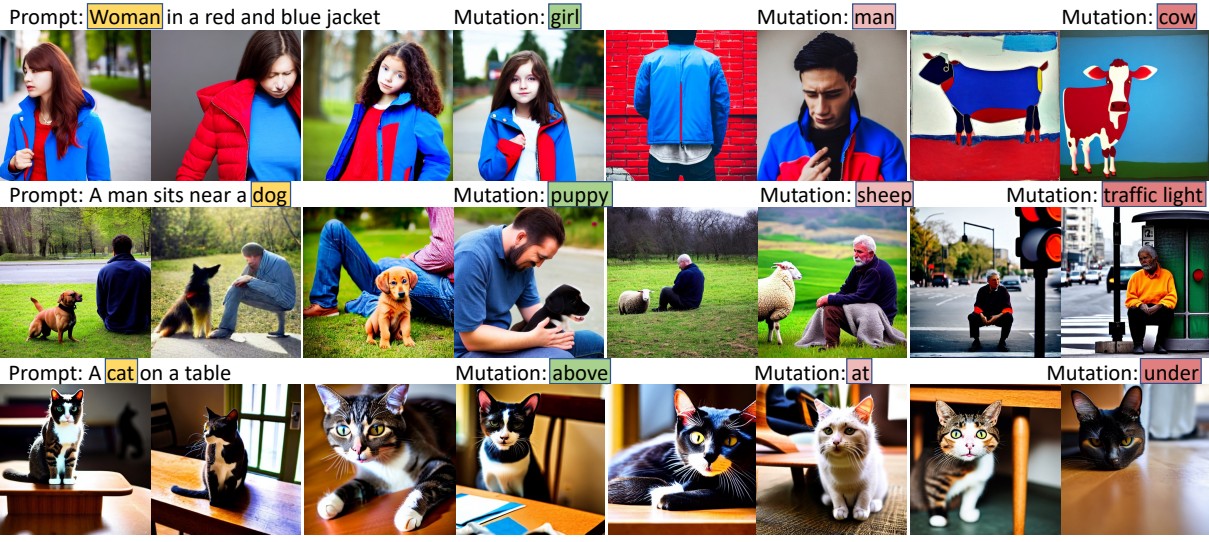

Figure 13: Visualization of the examples generated by Stable Diffusion with different mutations of semantic concepts in different prompts: the first row mutates the subject in the prompt, the second row shows the mutation of the object and the last row mutates the relationship between the subject and the object in the sentences.