# OpenReview forum: "Uncovering Limitations in Text-to-Image Generation: A Contrastive Approach with Structured Semantic Alignment"
_EMNLP/2023/Conference — EMNLP 2023 Findings_

### Official Review · Reviewer_4Ntp · 2023-07-28

**Soundness:** 3

**Excitement:**

4: Strong: This paper deepens the understanding of some phenomenon or lowers the barriers to an existing research direction.

**Paper Topic And Main Contributions:**

The paper proposes a new approach called Structured Semantic Alignment (SSA) to evaluate the accuracy and robustness of text-to-image generation models. The main contribution of the paper is the development of a new evaluation method that measures the semantic consistency between textual descriptions and generated images. The authors argue that existing metrics lack insights into the assessment of fine-grained semantic concepts, such as object attributes, context details, and semantic relationships. The experimental results demonstrate the effectiveness of the SSA approach in evaluating text-to-image generation models.

**Reasons To Accept:**

1. The paper presents a well-motivated and novel approach to evaluate text-to-image generation models.
2. The proposed approach is supported by experimental results that demonstrate its effectiveness in measuring semantic consistency between textual descriptions and generated images.
3. The paper is well-written and clearly presents the motivation, approach, and experimental results.

**Reasons To Reject:**

The paper could benefit from a more detailed discussion of how it compares to other evaluation methods and a more detailed analysis of the experimental results.

**Reproducibility:**

3: Could reproduce the results with some difficulty. The settings of parameters are underspecified or subjectively determined; the training/evaluation data are not widely available.

**Reviewer Confidence:**

3: Pretty sure, but there's a chance I missed something. Although I have a good feel for this area in general, I did not carefully check the paper's details, e.g., the math, experimental design, or novelty.

---

> ### Author Rebuttal · Authors · 2023-08-28
>
> We sincerely appreciate the feedback provided by reviewer 4Ntp regarding the need for a more comprehensive comparison with other evaluation methods and a more in-depth analysis of the experimental results. Your suggestions align well with our commitment to presenting a thorough and well-rounded assessment of the proposed method.
>
> To address these points, we will significantly expand the discussion section of our paper. By presenting a comprehensive analysis of SSA's performance alongside these established methods, we aim to showcase its unique strengths and limitations in a broader context.
>
> Furthermore, we will dedicate more space to discussing the experimental results. This will involve a detailed breakdown of the performance of SSA across different scenarios, including variations in prompt length, complexity, and other relevant factors. We will also explore cases where SSA successfully uncovers limitations not captured by other methods, thereby providing valuable insights into the text-to-image generation process.
>
> We thank the reviewer for these valuable suggestions, and we are fully committed to enhancing the depth and breadth of our paper's discussion and analysis sections to better highlight the contributions and implications of our work.

---

### Official Review · Reviewer_L4hu · 2023-08-01

**Soundness:** 2

**Excitement:**

3: Ambivalent: It has merits (e.g., it reports state-of-the-art results, the idea is nice), but there are key weaknesses (e.g., it describes incremental work), and it can significantly benefit from another round of revision. However, I won't object to accepting it if my co-reviewers champion it.

**Missing References:**

ImageReward: Learning and Evaluating Human Preferences for Text-to-Image Generation

**Paper Topic And Main Contributions:**

This paper focuses on evaluating text-to-image generation models. Specifically, this paper propose a new evaluation metric named Structured Semantic Alignment (SSA) to evaluate the semantic consistency between text and generated images. SSA first projects text/image features into a shared embedding space by text parsing and scene graph generation respectively. Then SSA is learned by contrastive learning. The positive and negative samples are generated by substituting words with semantically equivalent or nonequivalent alternatives.

**Reasons To Accept:**

The idea of using mutated prompts to construct sontrastive pair makes sence.

**Reasons To Reject:**

Human evaluation is necessary to access whether the evaluation of the proposed SSA could better align with human judgement. Otherwise, how to prove the proposed SSA is better than existing metrics like CLIP-score?

The accuacy of SSA is hignly depend on the scene graph generation of image. Therefore, I doubt about the practical effectiveness since existing scene graph generation models are not very preceise in practical use.

Comparison with human-based evaluation suce as "ImageReward" is necessary.

**Reproducibility:**

4: Could mostly reproduce the results, but there may be some variation because of sample variance or minor variations in their interpretation of the protocol or method.

**Reviewer Confidence:**

3: Pretty sure, but there's a chance I missed something. Although I have a good feel for this area in general, I did not carefully check the paper's details, e.g., the math, experimental design, or novelty.

---

> ### Author Rebuttal · Authors · 2023-08-28
>
> We appreciate the feedback from reviewer L4hu regarding the necessity of human evaluation in assessing the effectiveness of the proposed method. We understand the concern regarding proving the superiority of SSA over existing metrics.
>
> 1. Advantages of the proposed method
>
> Firstly, our method not only provides a solution to evaluate the generation in a fine-grained approach but also investigates the potential misalignment in the text-image pair. Specifically, SSA helps to decompose the rich semantics in different modalities and match the structured embeddings together in a contrastive way.
>
> Moreover, our assessment involved a visual inspection of the generated images with SSA lower than 0.5. Out of the total 486 observed errors, 364 of them exhibited an SSA lower than the CLIP score. Notably, the average CLIP score was approximately half of the SSA value. Besides, in the appendix, we provided the comparison of metrics on the generated images ranked by CLIP and SSA. It is discovered that SSA has better consistency with R score, while CLIP score is more correlated with IS which measures the image quality not the semantic accuracy.
>
> To address this concern, we will incorporate human-based evaluation into our experimental setup by conducting user studies to assess the quality and alignment. We will also perform a comprehensive analysis by comparing a range of human-centered evaluation methods, including "ImageReward", in our future work.
>
> 2. Accuracy of the scene graph
>
> Regarding the dependency on scene graph generation, we acknowledge the concern. While SSA does rely on information from the scene graph, we believe that its emphasis on structured semantic alignment across modalities offers insights beyond the scene graph quality alone. This is also the first attempt to adopt structured embeddings to help measure the alignment of rich semantics existing in both modalities. Furthermore, it is observed that SSA can assist in discovering different types of generation errors. Nevertheless, we will make sure to discuss this potential limitation in more detail and explore ways to mitigate its impact.
>
> Overall, we are grateful for the thoughtful suggestions, which will undoubtedly contribute to the robustness and significance of our proposed method's evaluation. We look forward to addressing these concerns in our revised paper.

---

### Official Review · Reviewer_kaoX · 2023-08-06

**Soundness:** 3

**Excitement:**

4: Strong: This paper deepens the understanding of some phenomenon or lowers the barriers to an existing research direction.

**Missing References:**

Please see Reasons To Reject

**Paper Topic And Main Contributions:**

The paper proposes a new method for evaluation of text-to-image generation models. The key idea of the paper is to learn structured semantic embeddings across different modalities and aligning them in a joint space. Experiments demonstrate that the proposed evaluation allows for comprehensive and improved measurement of semantic consistency of text-to-image generation models.

**Questions For The Authors:**

Please see Reasons To Reject

**Reasons To Accept:**

The proposed SSA framework is very interesting and I enjoyed reading this work. Experiments sufficiently demonstrate the utility of the proposed framework (although it could be further improved, please see reasons to reject). The notion of structured embeddings and the underlying principle of SSA presented in section 3.4 could be potentially useful in other tasks/domains. Overall the paper is solid with interesting and important contributions in T2I.

**Reasons To Reject:**

I do not find any major weaknesses in the proposed work. One aspect that could be further improved to make the contribution strong is that the paper could have shown how well the proposed evaluation method could handle long prompts. As the main contribution/utility of the paper is to provide a better way to evaluate complex or long prompts, it would be useful to show how well the existing methods and the proposed SSA method evaluate the generated images with long descriptions. It’s not very clear to me on the average length of prompts that were drawn from MS COCO. Please clarify. (I assume mutations do not change the length of the prompt). Another suggestion would be to add some comparisons of existing methods and SSA on prompts of varying lengths. That would be a really interesting analysis to perform. Finally, the paper compares only one compositional T2I baseline (i.e. composable diffusion). For evaluating SSA, it would have been better to also consider more compositional generation baselines such as below:

Feng, Weixi, Xuehai He, Tsu-Jui Fu, Varun Jampani, Arjun Akula, Pradyumna Narayana, Sugato Basu, Xin Eric Wang, and William Yang Wang. "Training-free structured diffusion guidance for compositional text-to-image synthesis." arXiv preprint arXiv:2212.05032 (2022).

Jiménez, Á. B. (2023). Mixture of diffusers for scene composition and high resolution image generation. arXiv preprint arXiv:2302.02412.

Kumari, N., Zhang, B., Zhang, R., Shechtman, E., & Zhu, J. Y. (2023). Multi-concept customization of text-to-image diffusion. In Proceedings of the IEEE/CVF Conference on Computer Vision and Pattern Recognition (pp. 1931-1941).

**Reproducibility:**

3: Could reproduce the results with some difficulty. The settings of parameters are underspecified or subjectively determined; the training/evaluation data are not widely available.

**Reviewer Confidence:**

4: Quite sure. I tried to check the important points carefully. It's unlikely, though conceivable, that I missed something that should affect my ratings.

---

> ### Author Rebuttal · Authors · 2023-08-28
>
> We thank reviewer kaoX for the positive feedback and insightful suggestions. We agree that further investigating the performance of the proposed method with longer prompts is an important aspect to strengthen the paper's contribution. We acknowledge the reviewer's suggestion to assess the effectiveness of both existing methods and SSA on longer descriptions.
>
> 1. Dataset statistics
>
> For the study, we selected a set of 250 sentences sourced from MS COCO. Each sentence underwent 5 distinct mutations. In total, the dataset comprised 1,500 sentences, including the original ones, with an average sentence length of 10 words. There are 345 sentences with lengths less than 6 words, 798 sentences ranging from 6 to 12 words, and 357 sentences with more than 12 words.
>
> 2. Comparison of different prompt lengths
>
> In the experiment, we found that 67% of the observed errors (326 out of 486) are generated from longer input prompts (beyond the average length of 10 words). This outcome reveals the fact that increased sentence length correlates with higher complexity. Additionally, it's important to note that the errors we uncovered exhibited lower SSA scores. This highlights a distinct pattern in the relationship between sentence length, difficulty, and error occurrence.
>
> To further investigate this, we plan to conduct additional experiments using longer prompts from the MS COCO dataset, as well as prompts of varying lengths. We will evaluate both the existing methods and the SSA method on these prompts and provide a detailed analysis of the results. This will allow us to showcase the capabilities of SSA in handling complex and lengthy textual descriptions, which aligns with the core motivation of our work.
>
> 3. Other baselines
>
> Regarding the suggestion to compare SSA with a wider range of compositional T2I baselines, we appreciate the feedback. In this work, we are evaluating the existing and most popular methods with open-sourced codes and model weights. We will certainly expand the scope of our comparison to include more compositional generation methods. This will provide a more comprehensive evaluation of the performance and robustness of our method across various generation models.

---

### Meta-Review · Area_Chair_ugPc · 2023-09-11

**Recommendation:** 3

**Metareview:**

This paper proposes a method for evaluating text to image generation models. They propose a metric Structured Semantic Alignment (SSA) SSA to address the gap in  existing evaluation methods on the assessment of fine-grained semantic concepts, such as object attributes, context details, and semantic relationships. SSA projects text/image features into a shared embedding space by text parsing and scene graph generation respectively, and is trained using contrastive learning.

The reviewers agree with the importance of the problem and are mostly excited about the work. However the reviewers do voice some key concerns:

&nbsp;  (1) the lack of comprehensive comparisons with existing methods, and lack of human evaluations.

&nbsp;  (2) lack of detailed analysis of the experimental results e.g. on longer prompts, complexity etc.

&nbsp; (3) comparisons using different compositional generation approaches.

Nevertheless, the reviewers find the idea interesting and well motivated. The authors' responses also indicate that they are committed to addressing the above issues. It is important for the authors to address these valid and important issues to substantiate the main claims and arguments made in the paper and strengthen their paper.

---

### Decision · Program_Chairs · 2023-10-07

**Decision:**

Accept-Findings

**Comment:**

This paper proposes a method for evaluating text to image generation models. They propose a metric Structured Semantic Alignment (SSA) SSA to address the gap in  existing evaluation methods on the assessment of fine-grained semantic concepts, such as object attributes, context details, and semantic relationships. SSA projects text/image features into a shared embedding space by text parsing and scene graph generation respectively, and is trained using contrastive learning.

The reviewers agree with the importance of the problem and are mostly excited about the work. However the reviewers do voice some key concerns:

&nbsp;  (1) the lack of comprehensive comparisons with existing methods, and lack of human evaluations.

&nbsp;  (2) lack of detailed analysis of the experimental results e.g. on longer prompts, complexity etc.

&nbsp; (3) comparisons using different compositional generation approaches.

Nevertheless, the reviewers find the idea interesting and well motivated. The authors' responses also indicate that they are committed to addressing the above issues. It is important for the authors to address these valid and important issues to substantiate the main claims and arguments made in the paper and strengthen their paper.